# The Journal of Open Humanities Data. Bridging open data and Wikidata for the Humanities

*Andrea Farina and Barbara McGillivray (King's College London)*

Wikidata serves as a critical tool for enriching and interconnecting datasets, enabling researchers to explore relationships across diverse domains (Farda-Sarbas and Müller-Birn 2019; Neubert 2017). It offers a centralized platform for integrating identifiers, metadata, and semantic links, and allows for the creation of interoperable and reusable datasets supporting advanced analysis and interdisciplinary research. At the Journal of Open Humanities Data (JOHD), we embrace the potential of Wikidata to amplify the impact of open data for research in the humanities. Through the publication of data papers (peer-reviewed articles describing datasets, their methodologies, and their reuse potential), JOHD ensures that humanities datasets are accessible and reusable. Our mission aligns with the principles of platforms like Wikidata, emphasizing transparency, accessibility, and collaboration to elevate the role of data in advancing scholarly work and public engagement (Wigdorowitz et al. 2024).

This poster highlights the synergies between JOHD and Wikidata, focusing on how the journal's principles of open access, reusability, and reproducibility complement Wikidata's capabilities as a linked open data hub. This collaboration can enhance the value and impact of humanities research in the digital age with JOHD acting as a bridge to encourage humanities scholars to engage with Wikidata by providing guidance on integrating datasets into Wikidata. We present case studies of data papers published in JOHD, showing how they have used Wikidata for dataset creation. For instance, linking place names in historical newspapers to Wikidata (Coll Ardanuy et al. 2022) enhances cultural heritage accessibility. Multilingual cultural heritage information, such as historical Chinese kung fu masters, can be integrated with Wikidata into reusable and human-centered knowledge graphs (Hou and Yuan 2023). Further, Wikidata ensures the reusability and transparency of bibliographical data, supporting JOHD's emphasis on reproducible research (Malínek et al. 2024). We also comment on published datasets that do not mention Wikidata but could potentially benefit from its integration to enhance interdisciplinarity (e.g., Farina 2023).

Finally, we explore how datasets published in JOHD and integrated with Wikidata can enhance the visibility and discoverability of research (cf. McGillivray et al. 2022) by tracking dataset reuse and citation within the Wikidata ecosystem.

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
