# OpenReview forum: "The Journal of Open Humanities Data. Bridging open data and Wikidata for the Humanities"
_wikimedia.it/Wikidata_and_Research/2025/Conference — WD&R Poster_

### Official Review · ~Rossana_Morriello1 · 2025-01-02
**Perfecly focused on the theme and practical application**

**Originality:** 5
**Impact:** 5
**Confidence:** 5

**Review:**

The use of Wikidata in scholarly journals is one of the most interesting fields of development of the collaboration between Wikidata and the academic community - and particularly in the humanities - which is the main topic of the conference. The aim of advancing public engagement, as stated in the proposal, is also extremely interesting and not very much considered in the use of data and datasets and so it is a quite original approach, which I would suggest to highlight.
The poster seems to be perfectly focused on the theme of the conference and presents a practical application of the use of Wikidata which could be a useful benchmarking opportunity for scholars and journal managers.

**Compliance:**

5

**Scientific Quality:**

5

---

### Official Review · ~Alessandra_Boccone1 · 2025-01-07
**Poster su vari argomenti chiave della conferenza**

**Originality:** 5
**Impact:** 5
**Confidence:** 5

**Review:**

Il poster mette in evidenza come le policy e i contenuti della rivista JOHD siano allineate con le possibilità offerte da uno strumento come Wikidata, che permette di scoprire relazioni fra domini diversi, con un'attenzione specifica al mondo dell'open access relativamente alle riviste scientifiche di ambito umanistico.  La presentazione tocca vari argomenti chiave della conferenza, come i metodi di gestione dei dati, le strategie di ricerca e la condivisione dei set di dati, di conseguenza si trova perfettamente in linea con ciò che è stato richiesto nel bando; anche la bibliografia a corredo è di qualità.

**Compliance:**

5

**Final Paper Review:**

In addition to the interesting content, I really appreciate the poster layout, clear and schematic. Excellent result!

**Scientific Quality:**

5

---

### Decision · Program_Chairs · 2025-02-05

**Decision:**

Accept (Poster)

**Comment:**

==Presence confirmed on february 11th==